# A UNIFIED EVALUATION FRAMEWORK FOR FROZEN VISUAL MODELS ON FORECASTING TASKS

## ABSTRACT

Forecasting future events is a fundamental capability for general-purpose systems that plan or act across different levels of abstraction. Yet, evaluating whether a forecast is "correct" remains challenging due to the inherent uncertainty of the future. We propose a unified evaluation framework for assessing the forecasting capabilities of frozen vision backbones across diverse tasks and abstraction levels. Rather than focusing on single time steps, our framework evaluates entire trajectories and incorporates distributional metrics that better capture the multimodal nature of future outcomes. Given a frozen vision model, we train latent diffusion models to forecast future features directly in its representation space, which are then decoded via lightweight, task-specific readouts. This enables consistent evaluation across a suite of diverse tasks while isolating the forecasting capacity of the backbone itself. We apply our framework to nine diverse vision models, spanning image and video pretraining, contrastive and generative objectives, and with or without language supervision, and evaluate them on four forecasting tasks, from low-level pixel predictions to high-level object motion. We find that forecasting performance strongly correlates with perceptual quality and that the forecasting abilities of video synthesis models are comparable or exceed those pretrained in masking regimes across all levels of abstraction. However, language supervision does not consistently improve forecasting. Notably, video-pretrained models consistently outperform image-based ones.

## 1 INTRODUCTION

The ability to see does not just reveal the present. It lets us anticipate the future and plan or act in the world accordingly. This capacity for visual forecasting is as critical for a gazelle dodging predators on the savanna as it is for a self-driving car navigating the urban jungle. At the same time, in most practical scenarios, the future is hard to predict. At any moment, countless possibilities lie ahead, and any model of the future must grapple with this inherent uncertainty.

While modern computer vision models learn representations general enough to work across multiple levels of abstraction such as DINOv2 Oquab et al. (2023) and 4DS Carreira et al. (2024), most focus on *perception* — tasks grounded in past and present frames with little to no stochasticity. Several methods of evaluations have been developed for self-supervised perception tasks, such as linear readouts, nearest-neighbors, cross-attention based, etc. However, evaluating vision models on these tasks tells us whether they understand what has already happened but may not reveal how well they can forecast what is to come.

In this paper, we shift the focus from evaluating the video *perception* capabilities of vision models to evaluating their video *forecasting* capabilities—assessing learned visual representations that can be used to predict future states of the world under uncertainty and across multiple levels of abstraction, such that diverse perceptually relevant quantities can be decoded from a single predictive representation.

We propose a unified forecasting evaluation framework built around a diffusion-based forecasting model, enabling forecasting across a range of frozen base video models and tasks: pixels, depth, point tracks, and bounding boxes. While recent work has explored using generative models for perception tasks (*e.g.*,Zhao et al. (2023); Luo et al. (2023); Li et al. (2023); Hedlin et al. (2023);

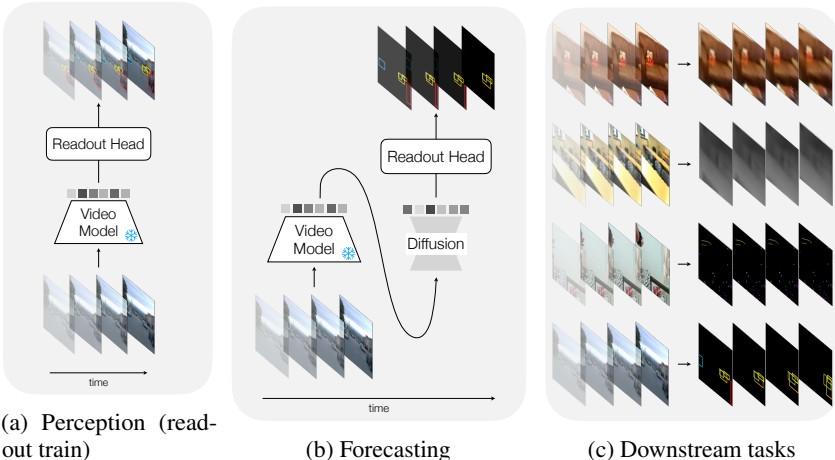

(a) Perception (readout train)

(b) Forecasting

(c) Downstream tasks

Figure 1: **Diffusion-based forecasting evaluation framework of frozen vision model backbones.** **(a)** *Perception-style readouts:* we train readout heads on frozen representations to perform downstream perception tasks like object detection on observed frames as in Carreira et al. (2024). We extend this setup to forecasting as follows. **(b)** *Forecasting framework:* We introduce a forecasting diffusion model that predicts future representations conditioned on frozen observed context representations. Pretrained readouts then decode these into future downstream abstractions, such as bounding boxes. **(c)** *Forecasting across abstraction levels:* We apply our approach to evaluate forecasting on tasks spanning low to high-level structure—pixels, depth, point tracks, and object detections. Each example shows 4 observed frames and a sample from the 12 forecast frames (frames 7, 10, 13, and 16). *Our results show that frozen video representations can generalize to forecasting across a wide range of downstream tasks.*

Zhang et al. (2023); Bhattad et al. (2023); Xu et al. (2025)), the reverse—using perception models for forecasting—has been less explored. We fill this gap by showing that frozen video models trained for perception *can* be effectively repurposed for forecasting, and we establish a benchmark and strong baselines to support future explorations.

Forecasting in video presents three key challenges. First, the future is inherently stochastic—multiple plausible outcomes can unfold from the same past. Second, forecasting is not about reaching a single endpoint, but about modeling how the future evolves over time as a continuous trajectory. Third, the future manifests at multiple semantic levels, from low-level pixels to mid-level motion tracks and high-level object abstractions. Our diffusion-based approach addresses all three: it captures uncertainty by generating diverse samples, models full temporal trajectories rather than single future states, and enables forecasting across a range of prediction targets, including pixels, depth, point tracks, and bounding boxes.

We extend frozen, pretrained state-of-the-art video models to forecasting tasks in two stages. First, we fit a lightweight attention-based readout head to each model for each downstream task, following the perception-based paradigm of Carreira et al. (2024) (Figure 1a). This readout head maps from the space of frozen video representations to task outputs (*e.g.* point tracks), trained using standard perception-based supervision. Then, we train a diffusion model to *forecast* future trajectories directly in the space of the frozen video representations (Figure 1b). During evaluation, we pass forecast trajectories through the readout head and assess their quality in the space of the downstream task (Figure 1c) using a suite of metrics that measure both realism and diversity—capturing the full dynamics and stochastic nature of the predicted futures.

Our diffusion-based forecasting framework enables direct, apples-to-apples comparison between perception- and synthesis-based models across all levels of visual abstraction. Our large-scale study reveals several key insights. First, forecasting ability is generally correlated with perception performance, but this association starts to break down with the strongest models. Second, video synthesis models like WALT Gupta et al. (2024) meet or exceed the forecasting performance of similarly-sized models trained with mask-based objectives when evaluated with a distribution-sensitive metric. Third, language supervision does not help forecasting performance. Fourth, models

trained solely on static images consistently underperform, highlighting the importance of temporal context in learning generalizable video representations.

## 2 RELATED WORK

**Video (pixel) synthesis.** Early video prediction models used recurrent architectures to directly model pixel intensities Ranzato et al. (2014); Oprea et al. (2022), but struggled with long-term dynamics. Probabilistic models like SVG-LP Denton and Fergus (2018) and GANs Clark et al. (2019); Tulyakov et al. (2018); Wang et al. (2020) improved visual quality by factoring content and motion. More recently, diffusion models Ho et al. (2020) have dominated video synthesis, with models like Sora OpenAI (2024), MovieGen Polyak et al. (2025), VideoPoet Kondratyuk et al. (2023) and WALT Gupta et al. (2024) leading the way. These models capture temporal dynamics through stochastic differential equations. While some diffusion-based works address video prediction Gu et al. (2023); Xing et al. (2024); Höppe et al. (2022); Ye and Bilodeau (2024); Yang et al. (2023), they are either language-guided or operate in implicit representation spaces. We leverage a diffusion model as a general forecasting engine across a range of visual abstractions. In addition, there has been work on diffusion video models which are autoregressive in time. Hu et al. (2024) is autoregressive on spatial-temporal blocks and leverages diffusion to generate unseen blocks. Zhang et al. (2025) and Huang et al. (2025) are autoregressive on past video frames and use diffusion to forecast the latents of future frames, similar to our proposed architecture. These approaches to autoregressive diffusion are potential design alternatives to the forecasting module in our evaluation framework, which would be interesting to explore in future work.

**Task-specific video-based forecasting.** Directly forecasting future pixels often fails to produce representations useful across abstraction levels. Luc et al. (2017) showed that forecasting semantic segmentation maps outperforms segmenting predicted RGB frames. They later proposed forecasting in the feature space of Mask R-CNN Luc et al. (2018), an approach similar to ours. Similarly, Vondrick et al. (2016) forecast features of AlexNet to predict actions and objects in the future. Similarly, other previous works Saric et al. (2020); Lin et al. (2021) also focus specifically on the task of forecasting segments or pixel interpolation Argaw and Kweon (2022). However, we generalize this framework: rather than relying on task-specific networks, we forecast in the frozen representation space of large pretrained video models that support a broad range of downstream tasks.

A separate line of work focuses on learning temporal dynamics from scratch using generative models or Neural Differential Equations (NDEs), such as Trajectory Flow Matching Zhang et al. (2024) and ImageFlowNet Liu et al. (2025). While these methods construct explicit, task-specific dynamical models, we ask how well general-purpose frozen video representations capture implicit dynamics that can be leveraged for forecasting via a separate diffusion-based module.

**Multi-task forecasting with frozen video representations.** Instead of designing task-specific forecasting models, several works have explored forecasting directly in the frozen representation space of large pretrained video models, leveraging their generality across downstream tasks. In this setup, future representations are predicted by learning lightweight forecasting heads on top of frozen features. Variants of this approach appear in recent work Rajasegaran et al. (2025); Karypidis et al. (2024). Rajasegaran et al. (2025) pretrain autoregressive models on large-scale video data and evaluate the resulting frozen features using probing tasks such as short-term interaction anticipation.

Most related to our work, DINO-Foresight Karypidis et al. (2024) and Back to the Features Baldassarre et al. (2025) forecast frozen DINOv2 Oquab et al. (2023) features using a masked transformer and autoregressive model, respectively, evaluating downstream tasks such as segmentation, depth, and surface normals at a single future time point. Unlike our approach, these works assume deterministic futures and perform single-step prediction, while we model uncertainty and evaluate the entire distribution of future trajectories.

**Stochastic approaches to forecasting.** In many contexts, visual forecasting is an inherently stochastic problem. A simple deterministic regressor may not necessary capture the full diversity of possible future outcomes. Some previous approaches have attempted to address this stochasticity. For example, Bhattacharyya et al. (2019) proposes a Bayesian model that jointly captures epistemic and observation aleatoric of future states. Makansi et al. (2020) uses mixture density networks to estimate

the location of objects like pedestrians and vehicles from an egocentric view. In this paper, we use a diffusion model to directly learn the continuous distribution of future features.

## 3   METHOD

While prior works have explored forecasting directly in pixel space, we hypothesize that latent spaces should be better because they make the scene structure more clear and remove non-semantic, hard-to-predict details, which should make prediction easier. Therefore, we start with frozen pretrained models, and use them to both represent the conditioning (past) video and the future video that we wish to predict. We develop a two-stage forecasting evaluation framework built around a diffusion-based forecasting module that operates directly in the space of frozen video representations. This setup allows us to extend representations trained for perception or pixel synthesis to forecasting tasks without fine-tuning. We first train lightweight readout heads to decode task-specific outputs from frozen representations. Then, we train a diffusion model to forecast future latent trajectories in the space of the frozen video representations. These forecast representations are passed through the same readouts, enabling the evaluation of nondeterministic futures across multiple semantic levels. The full pipeline is illustrated in Figure 1.

### 3.1   LATENT FORECASTING VIA DIFFUSION

We forecast future representations using a conditional denoising diffusion model Ho et al. (2020). Given a sequence of frozen representations up to time $t$, the diffusion model generates future latent trajectories for times $t + 1 \dots T$ conditioned on the past (Figure 1b). Unlike pixel-space synthesis, our model operates in the latent space of each frozen backbone, making it architecture-agnostic and capable of comparing perception and synthesis models under the same framework.

We train one diffusion forecasting module per frozen video model under consideration. In our experiments, we condition the diffusion model on the latent encodings of $t = 4$ past frames after applying layer normalization, as it was found that the diffusion model would occasionally struggle to forecast in unnormalized latent space. We model latent encodings of a $T = 16$ frame clip, which includes the 4 past frames and 12 future frames ($16 \times 224 \times 224 \times 3$ clip). The diffusion models the latent encodings of all these frames jointly in time. Even though the diffusion model may already have information on the first $t$ frames for conditioning, the entire clip is modeled jointly to account for temporally-entangled features.

The diffusion forecasting model never takes as conditioning any direct pixel information, only latent encodings from a given video model. This pipeline is the same whether the encoder is a video or image model. In the image encoder case, the latents are the stacked result of the image encoder on all frames.

### 3.2   TASK READOUT HEADS

We decode the sampled latent trajectories using the previously trained readout heads to evaluate forecast futures. This allows us to assess prediction quality across different abstraction levels using task-appropriate metrics (Figure 1c). We train a lightweight attention-based readout head to decode the output of each frozen model into a task-specific prediction space. We focus on four tasks that vary in their level of abstraction. These are pixels, depth, point tracks, and bounding boxes. The readout heads for the four tasks follow those of Carreira et al. (2024) with the architecture of the depth readout head also used (but trained separately) for the pixel readout task (Figure 1a). These readouts are trained using standard supervised losses and provide a shared interface for comparing models across different architectures and pretraining paradigms. Importantly, readout heads are trained only on observed (past and present) frames and remain fixed during forecasting. We train a readout head for each frozen video model and downstream task pair. During inference, the transformer-based readout head processes features from all frames, conditional and forecast, simultaneously via full attention. The loss from the readout heads is not backpropagated to the diffusion model.

## 3.3 Evaluation Metrics

We measure performance by evaluating the accuracy and realism of the entire future trajectory rather than a static single target timestep in the future. To do so, we use two perspectives to evaluation. The first is to measure performance on a *per example* basis, measuring the statistics (mean, variance, max, min) of task-specific metrics over a set of samples for each example. Because this uses task specific metrics, it gives a reference point versus the corresponding perception task. The second is on the *dataset* level, using Fréchet Distance and variance of samples from the ground truth dataset. We argue that these metrics best consider the fact that future forecasting is inherently a stochastic task.

**Per Example Metrics.** For each example, we take 10 samples from our diffusion model and report the statistics of task-specific metrics from the ground truth over each sample. For pixel prediction, we use PSNR. For depth prediction, mean absolute relative error. For point tracks, we use Jaccard Distance. For box tracking, we use intersection over union. In order to account for the stochastic nature of forecasting, we report mean, minimum, and maximum of per-example samples for the given metric. For relative comparison, we also report the perception, or standalone performance on the ground truth latents on all frames, of the readout heads alongside the per example metrics.

**Fréchet Distance.** In order to capture the inherent stochasticity of forecasting the future, we must allow for a *distribution* over possible future trajectories and ensure that this forecast distribution is similar to that of the target ground truth data. We therefore compute the Fréchet Distance (FD) Fréchet (1957), a distribution distance metric comparing the forecast versus the ground truth set distribution over trajectories, *in the output representation of each task*. While Ng et al. (2022) employed FD for motion forecasting evaluation and Thakkar et al. (2025) for self-driving cars, action, and object interactions, we extend the use of FD for evaluating forecasts of pixels, depth, point tracks, and object bounding box tracks. Specifically, we first represent forecasts as points in some fixed dimensional space. For point tracks and box tracks, we represent each trajectory as, respectively, a vector in a 24-dimensional (2D coordinates over 12 future frames) and 48-dimensional (4 coordinates over 12 frames) spaces. For depth and pixels, we downsample each of the 12 output frames to $14 \times 14$ patches, leading to a 2352-dimensional representation space. We then fit multivariate Gaussians to the predicted and ground truth distributions in this space, and compute the Fréchet distance Dowson and Landau (1982) between them. To ensure our output is always of a fixed size, we filter out trajectories that do not contain all the available data points (*e.g.* point tracks that are not visible across all target frames due to occlusion). Note that while Heusel et al. (2017); Unterthiner et al. (2019) compute FD in the Inception embedding space, there is no Inception here. We compute FD directly in the output representation of each downstream task. Explicit details are provided in Appendix A.1.

**Variance.** While FD considers the realism and stochasticity of the forecast futures, it is prudent to pair it with a measure of the variance over these futures to ensure that the forecast futures are as diverse as the ground truth ones. We report the variance of the trajectories over the temporal axis, averaged over all other dimensions. This specifically assesses whether methods always forecast static future trajectories, which may be realistic but certainly not diverse.

## 4 Experimental Setup

### 4.1 Downstream Tasks and Datasets

We center our evaluation on downstream forecasting tasks designed to span multiple levels of semantic abstraction, from raw pixels to high-level object bounding boxes. This diversity allows us to probe how well frozen video representations support different kinds of future prediction and identify where video models generalize, and where they fail. We visualize each of these tasks in Figure 1(c).

**Pixels.** We evaluate models on the task of forecasting future RGB frames in ScanNet Dai et al. (2017). While forecasting in pixel space is highly challenging and high-dimensional, it tests low-level generative fidelity and temporal coherence. Pixel forecasting captures fine-grained dynamics but is often sensitive to misalignment or visual ambiguity. We measure pixel accuracy using PSNR.

**Depth.** Predicting future depth maps tests a model's ability to reason about 3D scene geometry over time. It requires some abstraction beyond raw pixels while still relying on relatively dense spatial information. Depth forecasting is particularly useful for studying how models encode physical structure and motion. We measure mean absolute relative error in ScanNet.

**Point Tracks.** Forecasting the trajectories of dense visual features or tracked keypoints in the Perception Test dataset Pătrăucean et al. (2023). Point tracks offer a structured yet fine-grained measure of temporal consistency and motion understanding. Because the same points persist over time, they provide a strong signal for evaluating both representation quality and future modeling. We report Average Jaccard (Doersch et al. (2022)).

**Object Bounding Boxes.** Forecasting future object locations as bounding boxes focuses on semantic-level understanding of object motion and interaction. This task tests whether representations capture object permanence, affordance, and dynamics—crucial for robotics or autonomous driving applications. We report Mean Intersection over Union (IoU) on the Open Waymo dataset Sun et al. (2020).

## 4.2 BENCHMARKED MODELS

We benchmark a set of the highest performing and largest image and video models available. See the supplementary material for the model and pretrainig specs for all models under consideration.

**Image models.** We benchmark SigLIP-2B Zhai et al. (2023), a 2B-parameter vision transformer trained on image–text pairs using a contrastive binary classification objective, and DINOv2 Oquab et al. (2023), a 303M-parameter vision transformer trained purely on images using a self-distillation loss without any language supervision. Since these models are not natively trained on video, we follow Carreira et al. (2024) and append learnable temporal positional embeddings to their output features. This modification enables fair comparison with video models by allowing the readout heads to exploit temporal structure when trained on top of the frozen embeddings.

**Video models.** We evaluate two categories of video models. The first group consists of models trained using masking-based self-supervised objectives. VideoMAE Tong et al. (2022), VideoMAEv2 Wang et al. (2023), and 4DS-e Carreira et al. (2024) are trained to reconstruct masked pixels, while V-JEPA Bardes et al. (2024) uses a feature reconstruction loss based on predictions from a teacher network. VideoPrism Zhao et al. (2024) incorporates language supervision through contrastive learning between video and text during pretraining, followed by a second stage that applies a masked reconstruction loss on video. The second group includes WALT Gupta et al. (2024), a video synthesis model trained jointly for frame prediction by conditioning on a past-frames-based signal with a probability $p_{\text{fp}} = 0.1$. We leverage this built-in capability in a pipeline referred to as Native WALT (N-WALT). N-WALT is not a new model; it is simply the pretrained WALT model used exclusively in its forecasting mode. For this pipeline, a single forward pass is performed with the past-frames-based conditioning signal to extract predictive features from the model intermediate layers. These features are then directly decoded by lightweight readout heads to produce task-specific outputs, thereby obviating the need for a separate diffusion model. We use the same layers for feature extraction as in Vélez et al. (2025).

## 4.3 IMPLEMENTATION DETAILS

**Forecasting diffusion model and readout head.** Our diffusion implementation uses DDIM Song et al. (2021) and incorporates a cosine schedule Nichol and Dhariwal (2021). The underlying backbone denoiser is a vanilla 5-layer transformer. Each transformer layer employs multi-headed attention with 8 heads, utilizing 1024 total dimensions for queries, keys, and values, alongside a 2048-dimension hidden layer for the MLP. The training objective for the diffusion model minimizes the mean squared error between the denoised output of the model and the original latents, computed from a given video encoder. The diffusion model architecture is consistent across all of the underlying video models.

The training methodology for the task-specific readout heads is the same as in Carreira et al. (2024). Readout heads are attention based and trained with L2 error for pixels and depth. For point tracks, a weighted sum of Huber loss of positions and cross entropy over visibility and uncertainty is used. For box tracking, L2 loss between the labeled box coordinates and predicted position is used.

Because the base video model latents are frozen, the forecasting diffusion model and the readout head can be trained simultaneously. We found that layer normalization Ba et al. (2016) of the frozen latents is extremely important for forecasting performance. We train for 40k iterations at a batch size of 32 or an equivalent 160k iterations for a batch size of 4 for the memory intensive SigLIP. Aggregating across all experiments, we utilize approximately 144 days worth of tpu-v5 and v6 chips.

| Model | Pixels | | | Depth | | | Point Tracks | | | Box Tracks | | |
|---|---|---|---|---|---|---|---|---|---|---|---|---|
| | Mean ↑ | Best ↑ | FD ↓ | Mean ↓ | Best ↓ | FD ↓ | Mean ↑ | Best ↑ | FD ↓ | Mean ↑ | Best ↑ | FD ↓ |
| 4DS-e Reg. | 18.96 | 18.96 | 46.61 | 0.188 | 0.188 | 694.18 | 0.59 | 0.59 | 0.00070 | 0.58 | 0.58 | 2.26 |
| 4DS-e | 19.89 | 22.03 | 30.95 | 0.1937 | 0.096 | 533.0 | 0.58 | 0.61 | 0.00068 | 0.56 | 0.66 | 1.87 |
| WALT Reg. | 21.69 | 21.69 | 14.4 | 0.2196 | 0.2196 | 209.86 | 0.61 | 0.61 | 0.00140 | 0.54 | 0.54 | 2.44 |
| WALT 500M | 20.4 | 22.55 | 5.46 | 0.230 | 0.138 | 210.1 | 0.64 | 0.68 | 0.00134 | 0.50 | 0.58 | 2.47 |

Table 1: **A deterministic regressor may be optimal in predicting the mean outcome, but it fails to account for the variance in possible outcomes.** Comparison of forecasting with a deterministic regression model versus a stochastic diffusion model conditioned on 4 frames. Mean and Best represents the mean and best out of 10 samples (for diffusion) on the task specific metric. For regression, there is only 1 deterministic output. For pixels, this is PSNR. For depth, Mean Absolute Relative Error. For point tracks, it is Jaccard Distance. For Box Tracks, it's IoU. FD is Frechet Distance in the output space.

**Evaluation protocol.** For pixels and depth, we use the standard train and validation split on ScanNet Dai et al. (2017). For point tracks, we train in the Kubric movie dataset Greff et al. (2022) and test on the Perception Test dataset Pătrăucean et al. (2023). On box tracking, we use the train and validation split of the Waymo Open Box dataset Sun et al. (2020). For all forecasting tasks we take as context 4 frames and forecast the next 12 frames in all experiments. We sample the diffusion model 10 times per example during evaluation. We use TPUs ranging from 4-32 chips (v5p and v6e) and report total training time for the experiments in Table 7.

**WALT setup.** We utilized WALT, a text-to-video diffusion model, as a frozen encoder, probing its intermediate layers in a setup similar to Vélez et al. (2025). WALT is designed to process 17 video frames, tokenizing them into five latent representations: one for the initial frame and four for the subsequent 16 frames. To maintain consistency with other models in this study, we sampled 16 frames, duplicated the first frame, and simulated the forward diffusion process by adding noise at timestep $t$. Instead of the complete multi-step generative process, the visual representation is obtained by a single forward pass through the denoiser, utilizing a null text embedding. During the single pass, the intermediate representations are extracted, discarding the initial latent representation. We use the same layers for feature extraction as in Vélez et al. (2025).

# 5 RESULTS

**The need for a stochastic evaluation.** To demonstrate the need for modeling the stochasticity of future events, we perform an ablation of our proposed diffusion forecasting module and compare it to regression-based forecasting in Table 1. We find that while regression optimizes the traditional mean-based metric, this does not account for the inherent stochasticity of forecasting. When considering metrics such as Fréchet Distance or Best-of-N that take this stochasticity into account, we find that diffusion models generally outperform the regression baselines in most cases.

**Forecasting mostly correlates with perception.** Overall, we observe a strong correlation between per-example metrics and the perception performance in Figure 2 where forecasting performance is displayed alongside perception performance. A more nuanced picture emerges when looking at the table in more detail. Here we find that, with the exception of box tracking, the best model in perception for a given task is not the best model for forecasting. At higher levels of performance, it appears that the dynamics of certain representations are inherently easier to model than others even if other representations contain more information relevant to the downstream task. In addition, we see that none of the representations we tested are universally optimal for both forecasting and perception on all tasks. We note that in Figure 2, Walt is the best on pixel tracks, Native WALT on pixels, 4DS-e on box tracks, and DinoV2 on Depth. In Figure 2, WALT excels on pixels and depth while VideoMAE excels on pixel tracks, and 4DS-e excels on box tracks. These results imply that specific models, while not necessarily optimal for perception on a given task, are optimal for forecasting for that specific task.

**Synthesis models like WALT achieve forecasting performance on par with or better than models trained with mask-based objectives.** WALT significantly excels at pixel and depth fore-

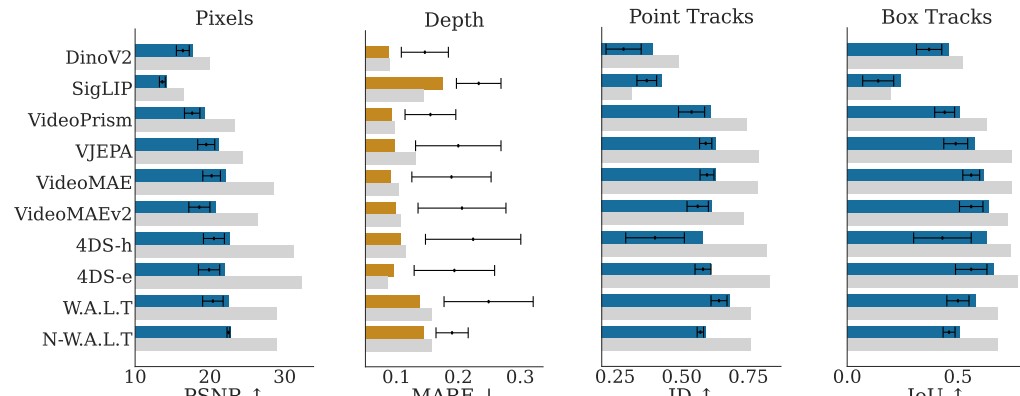

Figure 2: **Forecasting per-example metric results.** We evaluate forecasting on pixels (PSNR), point tracks (Jaccard Distance), bounding box tracks (IoU), and depth maps (Mean Absolute Relative Error) using 10 samples per example. The colored bars represent the best of n metrics for a particular task. Blue represents tasks where higher performance is better, while gold represents a metric where lower is better. We also report perception performance on each task as gray bars. Given the stochastic nature of forecasting, we report the **mean** (as whisker plot with standard deviation) and **maximum/minimum** performance (colored bars) across samples. This reveals differences in sample quality not captured by the mean alone—some models exhibit similar averages but differ significantly in their best-case outputs, highlighting variation in their predictive distributions. *Overall, we observe that stronger perception models tend to yield better forecasting performance. However, with the exception of box tracking, the best model in forecasting is never the best in perception.*

| Task (Dataset) | Pixels (ScanNet) | | Depth (ScanNet) | | Points (Perc. Test) | | Boxes (Waymo) | |
|---|---|---|---|---|---|---|---|---|
| | FD↓ | Var.($10^{-3}$) | FD↓ | Var.($10^{-3}$) | FD↓($10^{-3}$) | Var. | FD↓ | Var. |
| *GT* | | *12.00* | | *193* | | *0.039* | | *0.0032* |
| DINOv2 | 62.97 | 3.1 | 588.36 | 8.9 | 1.9 | 0.038 | 3.08 | 0.044 |
| SigLIP | 203.32 | 0.5 | 849.18 | 2.4 | 3.0 | 0.038 | 3.22 | 0.05 |
| VideoPrism | 70.30 | 6 | 882.11 | 7.3 | 0.8 | 0.039 | 2.72 | 0.045 |
| VJEPA | 37.08 | 5.1 | 558.28 | 7.7 | 0.63 | 0.039 | 2.85 | 0.048 |
| VideoMAE | 28.14 | 7 | 547.40 | 9.2 | 0.55 | 0.039 | 2.62 | 0.045 |
| VideoMAEv2 | 33.75 | 5.9 | 578.51 | 7.7 | 0.74 | 0.039 | 2.92 | 0.048 |
| 4DS-h | 28.29 | 10.0 | 555.11 | 8.5 | 7.88 | 0.039 | 3.24 | 0.054 |
| 4DS-e | 30.95 | 6.2 | 533.00 | 11.0 | 0.68 | 0.039 | 1.87 | 0.036 |
| WALT 500M | 5.46 | 6.9 | 210.10 | 5.6 | 1.34 | 0.039 | 2.47 | 0.040 |
| N-WALT 500M | 6.55 | 5.4 | 217.8 | 4.3 | 1.39 | 0.038 | 3.23 | 0.055 |

Table 2: **Distributional alignment of forecast futures.** We report Fréchet Distance (lower is better) and the variance of fitted Gaussian distributions for each metric, comparing the ground truth distribution to the model's sampled forecasts. Ideally, the forecast variance should closely match the ground truth. Results show that *stronger perception models produce forecasts with distributions more aligned to the data*, reinforcing trends observed in the per-example metrics (Fig. 2).

casting—tasks closely aligned with its training objective—as revealed by FD when evaluating against masked representation-learning models of similar size (VideoMAE and 4DS-h). WALT also outperforms 4DS-h in both point and bounding box forecasting, while performing comparably to VideoMAE in these tasks. This outcome does not align with the lower perception performance of WALT when benchmarked against these two models for depth prediction and object tracking. It is worth noting that N-WALT does not exhibit the same performance. Since it was trained with a frame prediction objective and is conditioned on past frames, it excels at pixel forecasting, effectively capturing low-level spatiotemporal dynamics. However, it underperforms in other tasks that require higher-level semantic understanding, such as point and box tracking. This performance disparity reveals a fundamental limitation of the pixel prediction objective, suggesting that the learned features are not truly generalizable.

**Vision-language contrastive training does not result in better forecasting.** In Table 2, we find that models with vision-language contrastive supervision like SigLIP and VideoPrism, which were trained only on perception-style tasks, lag behind.

**Video backbones outperform image ones.** Notably, we find that models pretrained exclusively on image-based objectives, such as DINOv2 and SigLIP, perform poorly across most tasks, reinforcing the importance of temporal supervision during pretraining, contrary to widespread belief Baldassarre et al. (2025). We note that the predictor model in Baldassarre et al. (2025) also uses past temporal context to forecast the future. However, Baldassarre et al. (2025) claims that the base visual representation on top of which this temporally-aware predictor operates can simply be image-based, while we demonstrate experimentally that base models trained on video outperform those trained on images alone. One notable exceptions to this finding is DINO which excels in depth forecasting, perhaps because ScanNet, the dataset we use in our evaluation for depth has limited motion and thus requires less dynamics information for predicting the future.

**Per-example vs. distribution-level metrics.** To compare our two proposed metric approaches, we first observe the per-example forecasting metrics in Figure 2. Unsurprisingly N-WALT is the strongest in forecasting pixels, given it was directly trained to do so. However, for depth forecasting, DinoV2 seems to exhibit the best per-example results. WALT with the trained diffusion head, but not N-WALT, is the best on forecasting point tracks. Interestingly, VideoMAEv2 underperforms its predecessor across nearly all metrics, while VideoMAE (v1) shows strong results in bounding box forecasting (per-example) and point tracks (FD), despite its smaller model size.

We next turn to the distribution level metrics in Table 2. Overall, we observe a strong correlation between the per-example metrics in Figure 2 and the distributional alignment of predicted futures with ground truth, as captured by FD and variance. However, we also find notable discrepancies emerge between the two forecasting evaluation paradigms. For instance, in depth forecasting, WALT performs strongly in terms of Frechet Distance, but DinoV2 seems to exhibit better per-example results. Similarly WALT excels on per-example metrics for point tracking, yet VideoMAE is better in terms of Frechet Distance on this task. This discrepancy highlights how small-sample evaluations can obscure poor distributional alignment.

We find that all models on most tasks, with the exception of point tracking, struggle to approach the variance of the ground truth datasets. Even though WALT performs relatively well on pixel forecasting with respect to the Frechet Distance metric, it still does not capture the full extent of the underlying variance in pixel space. The variance disparity is especially apparent with depth forecasting; this suggests that current models particularly struggle to model visual information relevant to this particular domain.

The N-WALT results reported in Table 2 and Figure 2 reflect performance at the specific noise levels that yielded the best metric values. The metrics across various noise levels are presented in Tables 5 and 6, with the results suggesting that maintaining low noise levels is critical for achieving the best results across tasks.

**Qualitative.** We visualize forecasts from the 4DS-e model in Figure 3. These results demonstrate that our approach generalizes well across multiple forecasting domains and abstraction levels.

**Ablations.** We ablated the effect of the readout head capacity as well as the number of samples taken from the diffusion model. These results are shown in Tables 9 and 8 respectively. We find that our results are do not change dramatically with respect to these variables.

## 6 DISCUSSION

We proposed a unified evaluation framework of frozen vision backbone models in forecasting tasks. Our central findings are first, that forecasting performance in frozen pretrained video models closely tracks their perception performance up to a point. At higher levels of performance, this association starts to break down. Second, as expected, WALT, a video synthesis model explicitly trained to generate future frames, significantly outperforms masked video models on low-level forecasting tasks such as pixel and depth prediction. However, WALT's forecasting strength is mixed for mid-level structured tasks like point tracks and object bounding boxes when comparing masked models of similar size. Third, language supervision alone does not appear to improve forecasting ability,

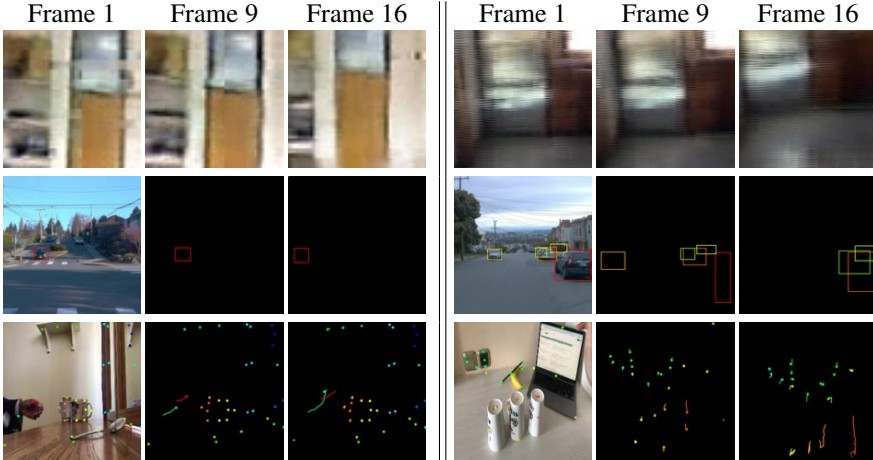

| Frame 1 | Frame 9 | Frame 16 | Frame 1 | Frame 9 | Frame 16 |

Figure 3: **Qualitative forecasts from the 4DS-e model across diverse tasks.** We condition on frames 1–4 and forecast frames 5–16. **Top:** Pixels forecasting—the model captures smooth camera motion. **Middle:** Bounding boxes—it predicts a car turning (left) and vehicle motion (right). **Bottom:** Point tracks—the model forecasts a hand rising (left) and camera motion (right). These results demonstrate that our approach generalizes well across forecasting domains and abstraction levels.

underscoring the importance of temporal visual learning for anticipating future states. Lastly, our results clearly show that video backbone models outperform their image-based counterparts in supporting future-forecasting tasks.

**Limitations.** Our forecasting evaluation framework has a few key limitations. First, the datasets studied, while diverse, often lack complex or ambiguous motion, limiting the generality of the tasks. Second, the introduction of a diffusion model is a potential confounding variable in evaluating the forecasting capabilities of frozen models. To mitigate this risk, we opt to use a simple "vanilla" transformer to provide a standardized and simple forecasting module while preventing an overly powerful, task-specific forecaster from compensating for weaknesses in a backbone's representations. We note that while diffusion provides a unified and expressive forecasting mechanism, it is computationally expensive and sensitive to sample count. Third, forecasting on frozen perception features may impose representational mismatches, especially for tasks requiring fine-grained temporal dynamics. Fourth, it is difficult to perfectly control for factors such as the number of model parameters, resolution of feature spaces, and the pre-training data volume for each frozen pretrained model. Lastly, we use video models with 16-frame contexts, which restricts our ability to assess long-horizon forecasting. That said, our evaluation framework is generic in that it will apply in exactly the same way to longer-context models, once they become available.

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
