# OpenReview forum: "A Unified Evaluation Framework for Frozen Visual Models on Forecasting Tasks"
_ICLR.cc/2026/Conference — Submitted to ICLR 2026_

### Official Review · Reviewer_432F · 2025-10-15

**Soundness:** 3
**Presentation:** 3
**Contribution:** 3
**Rating:** 6
**Confidence:** 3

**Summary:**

This paper introduces a novel, unified framework to efficiently assess the predictive power of frozen vision backbones across various levels of abstraction, from pixel synthesis to object bounding box tracking. The core method employs a lightweight Diffusion Model trained to forecast future trajectories directly within the frozen model's feature space, circumventing the need for expensive fine-tuning. Critically, the framework moves beyond traditional deterministic errors by emphasizing stochasticity, using distributional metrics like Fréchet Distance to accurately capture the multimodal and uncertain nature of future outcomes. The key findings confirm that video pre-training is essential for superior forecasting and reveal that powerful video synthesis models, such as WALT, possess unexpectedly strong predictive capabilities.

**Strengths:**

1. I believe the overall framework design is sound, appearing both effective and convenient, thus advancing the methodology for evaluating the quality of video representations.
2. The comparative evaluation of various pre-training strategies (e.g., language supervision, masked modeling) provides clear empirical guidance for building high-performance visual forecasting systems.
3. It proposes a unified evaluation framework capable of assessing diverse forecasting tasks across different levels of abstraction (from pixels and depth maps to point tracks and object bounding boxes) within a single architecture.

**Weaknesses:**

1. If there are image or depth prediction methods as a comparison (it should be easy to add some CNN-based or diffusion-based solutions from recent years), we can better understand the current level of this evaluation framework.
2. I'm concerned that the lightweight readout head could become a performance bottleneck, and it would be nice to have some experiments to illustrate the architectural choices for the head.

**Questions:**

* Expect additional experiments on the above weakness
* The framework uses Forecasting Future (FFF) performance to evaluate the quality of a frozen visual model's representation. Beyond the observed correlation, what is the theoretical or empirical justification for FFF being a reliable and essential proxy for assessing the fundamental quality of a visual representation itself?
* The forecasting capability relies on training an additional Diffusion Model in the latent space. How can the authors definitively prove that the measured FFF performance is not primarily bounded by or biased towards this newly trained Diffusion Model, rather than accurately reflecting the intrinsic predictive potential of the frozen backbone's features?
* The paper correctly highlights the "inherent uncertainty of the future." However, the final evaluation still relies on comparing predictions (via FD or Best-of-N) against the Ground Truth (GT) dataset labels. Does this approach truly solve the claimed difficulty of forecasting, or does it merely provide a more scientific way to compare against a known outcome? Have the authors considered or explored GT-independent intrinsic metrics for evaluating future plausibility?

---

> ### Author Response · Authors · 2025-11-21
>
> We would like to thank you for your helpful comments. You note that the “framework design is sound,” the framework provides “clear empirical guidance for building high-performance visual forecasting systems,” and proposes a “unified evaluation framework … within a single architecture.” We address your comments and questions below:
>
> **If there are image or depth prediction methods as a comparison (it should be easy to add some CNN-based or diffusion-based solutions from recent years), we can better understand the current level of this evaluation framework.**
>
> We will run experiments and update the submission with results to compare against image and depth prediction approaches.
>
> **I'm concerned that the lightweight readout head could become a performance bottleneck, and it would be nice to have some experiments to illustrate the architectural choices for the head.**
>
> We will run experiments to ablate the capacity of the readout head and will update the submission with results.
>
> **The framework uses Forecasting Future (FFF) performance to evaluate the quality of a frozen visual model's representation. Beyond the observed correlation, what is the theoretical or empirical justification for FFF being a reliable and essential proxy for assessing the fundamental quality of a visual representation itself?**
>
> We argue that our paper is in the spirit of William Gibson’s statement, "We see in order to move; we move in order to see". In natural forms of intelligence, there is no use for perception without action. In addition, we most often than not rely on forecasting future events from perceived information. Therefore, the ultimate test for visual representations should really be their usefulness in forecasting the future. Previous studies have shed only very limited light on the abilities of visual representations on forecasting tasks. This paper attempts to take an initial step in that direction and asks how well we can forecast with existing strong pre-trained representations. Is it the case that we need to train new ones for forecasting from scratch with novel training objectives?
>
> **The forecasting capability relies on training an additional Diffusion Model in the latent space. How can the authors definitively prove that the measured FFF performance is not primarily bounded by or biased towards this newly trained Diffusion Model, rather than accurately reflecting the intrinsic predictive potential of the frozen backbone's features?**
>
> This is a good question. While it may be unclear if diffusion models reflect the full predictive potential of the features, the models we use are all trained with the same diffusion architecture. Our conclusions are based on the relative performance of different models within our consistent evaluation pipeline. Because of this consistency, we argue that the variability we observe is likely due to the underlying representations. Additionally, we touch upon this somewhat using regression vs diffusion for forecasting on with some representations (such as WALT and 4DS). While perhaps not conclusive, the results do suggest that predictive potential seems to hold across different approaches to forecasting (regression vs diffusion)
>
> **The paper correctly highlights the "inherent uncertainty of the future." However, the final evaluation still relies on comparing predictions (via FD or Best-of-N) against the Ground Truth (GT) dataset labels. Does this approach truly solve the claimed difficulty of forecasting, or does it merely provide a more scientific way to compare against a known outcome? Have the authors considered or explored GT-independent intrinsic metrics for evaluating future plausibility?**
>
> Our proposed distribution-based metrics are designed specifically to compare the distribution of the forecasted prediction to the distribution of the ground truth data. This allows us precisely to accommodate the possibility of multiple data modes due to the inherent uncertainty of the future. Instead of the usual paradigm of comparing each forecasted sample to a single ground truth which ignores uncertainty, we allow for the fact that taken as a whole, the ground truth data should exhibit the multiple possible modes of the future. Naturally, we assume that the ground truth is diverse enough, such that the GT distribution is close to the real distribution over future states.

---

> > ### Comment · Reviewer_432F · 2025-11-27
> >
> > I have read the other comments and the rebuttal. I believe the authors need more detailed experiments and a more comprehensive evaluation to change the impression of the reviewers who gave low scores.
> >
> > For me, the current rebuttal also lacks sufficient detail.

---

> ### Author Response · Authors · 2025-12-03
> **Experiments**
>
> 1. We are running a comparison with DINO-Foresight on the Scannet dataset. However, as this dataset is much larger than the Cityscapes dataset in the original setup of DINO-Foresight, this experiment is still running and will not finish by the end of the rebuttal period. We will update the camera ready submission with this result.
>
> 2. Here is a table of an ablation of the 4DS-e model with different capacities of the transformer-based readout head. We find that varying the capacity (at least within these ranges) does not dramatically affect performance.
>
> | Model | Pixels Perception | Pixels Mean $\uparrow$ | Pixels Best $\uparrow$ | Pixels FD $\downarrow$ | Depth Perception | Depth Mean $\downarrow$ | Depth Best $\downarrow$ | Depth FD |
> | :--- | :---: | :---: | :---: | :---: | :---: | :---: | :---: | :---: |
> | 4DS-e 512 | 29.99 | 20.40 | 22.39 | 29.74 | 0.087 | 0.187 | 0.096 | 527.72 |
> | 4DS-e 1024 | 32.26 | 19.89 | 22.03 | 30.95 | 0.086 | 0.194 | 0.096 | 533.00 |
> | 4DS-e 2048 | 31.62 | 20.50 | 22.58 | 30.46 | 0.088 | 0.192 | 0.093 | 530.64 |
>
> **Ablation of performance at different capacities of the transformer-based readout head.** 1024 is the default number used in the paper.

---

### Official Review · Reviewer_X1vx · 2025-10-31

**Soundness:** 2
**Presentation:** 2
**Contribution:** 1
**Rating:** 2
**Confidence:** 3

**Summary:**

The proposed framework for Frozen Visual Models on Forecasting Tasks
 evaluates entire trajectories and incorporates distributional metrics that better capture
the multimodal nature of future outcomes. Given a frozen vision model, latent diffusion models are trained
to forecast future features directly and then decoded via lightweight, task-specific readouts

The framework is evaluated using 9 diverse vision models, spanning image
and video pretraining, contrastive and generative objectives, and with or without
language supervision, over 4 forecasting tasks, from low-level pixel predictions to high-level object motion.

The authors state that language supervision does not consistently improve forecasting.
Notably, video-pretrained models consistently outperform image-based ones.

**Strengths:**

Short and compact literature review

Brief discussions on proposed work - split into two  main parts:
LATENT FORECASTING VIA DIFFUSION
&
TASK READOUT HEADS.

Compact illustration of fig. 1 gives a brief idea of the proposed model.

Few tabular results shown.

**Weaknesses:**

Not  a single equation/expression with analytics presented.

Even if the work is quite intensive, it appears as a technical report for evaluation of the model.

There is no mention of contribution or novelty in the paper.

The overall framework seems to be dependent/derived from :
 - conditional denoising diffusion model Ho et al. (2020)
and
 - the readout heads for tasks as used in Carreira et al. (2024).

The GPU architecture platform used for training/testing is also not mentioned.

Effect of dataset bias during training may be highlighted.

The dark background in few image samples on Fig. 3, makes it difficult to comprehend - what authors want to highlight/exhibit.

**Questions:**

How does your proposed method handle uncertainties & sharp changes ?

What range of resolution of the frames do you method deal with?

What are the failure cases ?

---

> ### Author Response · Authors · 2025-11-21
>
> We thank you for your insightful comments. We address comments below:
>
> **“There is no mention of contribution or novelty in the paper.”**
>
> We discuss contributions on lines 100-109. We have a number of insights:
> - Forecasting ability is generally correlated with perception performance, but this association starts to break down with the strongest models.
> - Video synthesis models like WALT Gupta et al. (2024) meet or exceed the forecasting performance of similarly-sized models trained with mask-based objectives when evaluated with a distribution sensitive metric.
> - Language supervision does not help forecasting performance.
> - Models trained solely on static images consistently underperform, highlighting the importance of temporal context in learning generalizable video representations.
>
> **The overall framework seems to be dependent/derived from :conditional denoising diffusion model Ho et al. (2020) and
> the readout heads for tasks as used in Carreira et al. (2024).**
>
> We do indeed use denoising diffusion following the foundational work of Ho et al. However, our specific contribution lies in adapting this framework to vision-based forecasting tasks, while using pre-trained video perception models. Ours is the first work to introduce this adaptation. Regarding Carreira et al., we do use these readouts, but in this context we extend their use for forecasting.
>
> **The GPU architecture platform used for training/testing is also not mentioned.**
> **Effect of dataset bias during training may be highlighted.**
>
> We will update the submission with a discussion of these points.
>
> **How does your proposed method handle uncertainties & sharp changes ?**
>
> Our diffusion-based approach captures uncertainty by generating diverse samples, and our proposed metrics such as Frechet Distance and best of N samples account for the stochastic nature of forecasting. The datasets used in our evaluation generally do not have sharp changes.
>
> **What range of resolution of the frames do you method deal with?**
>
> We use a resolution of 224x224.
>
> **What are the failure cases ?**
>
> We discuss limitations in the appendix. We can update the submission to include this in the main paper.
>
> Our forecasting evaluation framework has a few key limitations.
> - The datasets studied, while diverse, often lack complex or ambiguous motion, limiting the generality of the tasks.
> - The introduction of a diffusion model is a potential confounding variable in evaluating the forecasting capabilities of frozen models.
> - Forecasting on frozen perception features may impose representational mismatches, especially for tasks requiring fine-grained temporal dynamics.
> - We use video models with 16-frame contexts, which restricts our ability to assess long-horizon forecasting.

---

> > ### Author Response · Authors · 2025-12-03
> > **Hardware Used**
> >
> > We use TPUs ranging from 4-32 chips (v5p and v6e), and we discuss issues related to dataset bias in the limitations section. We have updated the submission with a discussion of these points.

---

### Official Review · Reviewer_JPXb · 2025-11-01

**Soundness:** 1
**Presentation:** 2
**Contribution:** 2
**Rating:** 2
**Confidence:** 3

**Summary:**

1. The authors introduce a method to evaluate frozen vision models on stochastic forecasting tasks by training latent diffusion models to predict future representations which are decoded by task specific readout heads to predict future states at different abstraction levels from points to boxes.

2. The authors use this proposed evaluation framework to benchmark several popular frozen visual models with different pretraining strategies on their effectiveness at forecasting at different levels of abstraction.

3. Based on this evaluation, the authors present some insights/claims about the frozen vision models.

**Strengths:**

1. The authors introduce a novel evaluation framework based on diffusion that can benchmark frozen vision models on inherently stochastic forecasting tasks at different levels of abstraction.

2. The authors use distribution evaluation metrics like FID and variance to measure the diversity and realism of the predicted future states.

3. The authors run extensive experiments to benchmark ~10 frozen vision models with different pretraining strategies on 4 types of tasks.

**Weaknesses:**

1. The analysis done by the authors does not provide any interesting insights:

-  **"Forecasting mostly correlates with perception"** this is not surprising at all, rather it is expected that better perception models will generally also be better at forecasting. Unfortunately, the one somewhat interesting finding here "the best model in perception for a given task is not the best model for forecasting" is not investigated in detail by the authors.
- **"Synthesis models like WALT achieve forecasting performance on par with or better than models trained with mask-based objectives"** This can very likely be attributed to the fact that WALT is a diffusion model, meaning its diffusion representations will likely be better suited for a diffusion based forecasting module. The authors should investigate this in detail, preferably using a non-diffusion based synthesis model. If no such open-source model exists for videos that is comparable in size and scale to other frozen video models, then this ablation can be done for image synthesis models.
- **" N-WALT does not exhibit the same performance"** It is highly likely that this discrepancy arises out of the fact that the authors perform  a single forward pass to get the intermediate features from a video diffusion model. Diffusion models are supposed to take different noise levels as input, with high noise levels capturing broad semantics and low noise levels capturing high fidelity details. So a single forward pass, (likely done at a fixed low noise level) is going to be better at low level pixel forecasting and worse at semantic box tracking. The authors should try different noise levels and evaluate again.
- **"Language supervision does not result in better forecasting"** It is not accurate to make this claim, since WALT is in itself a text to video model, pre-trained with text supervision. Maybe the authors mean contrastive here.
- **"Video backbones outperform image ones"** All things being equal we expect video models with temporal modelling capabilities to outperform image models with no such capacity at forecasting. The cited paper here, DINO-world, does not support the authors' claim since DINO-world does cross attention on past frame representations to learn the temporal modelling capacity, crucial for future prediction. So in the absence of this capacity in the image models, the temporal modelling is relegated to the latent diffusion module introduced by the authors which is the same for all vision models. This means the video models with inherent temporal modelling ability have the edge over image models in this case.

2. The authors also do not control for resolution of feature spaces or model params or pre-training data volume in their analysis. But this maybe excused given the difficulty of such a thing with pre-trained models all trained differently. But the authors should also address this concern in detail.

**Questions:**

see weaknesses

---

> ### Author Response · Authors · 2025-11-21
>
> We would like to thank you for your insightful comments. We are delighted that you found our approach to be a “novel evaluation framework,” the metrics proposed “measure the diversity and realism of the predicted future states,” and our evaluation is “extensive.” We address your comments below:
>
> **"Forecasting mostly correlates with perception"**
>
> While it is true that forecasting generally correlates with perception, we emphasize (e.g. on line 104-105) that there is more nuance to this result; this association starts to break down with the strongest models. We experimentally find that none of the representations we tested is universally optimal for both forecasting on all tasks. We note that in Figure 2, Walt is the best on pixel tracks, N-WALT on pixels, 4DS-e on box tracks, DinoV2 on Depth. In Table 2, WALT excels on pixels and depth while VideoMAE excels on pixel tracks, and 4DS-e excels on box tracks. These results imply that specific models, while not necessarily optimal for perception on a given task, are optimal for forecasting for that specific task. We will clarify this finding in the submission by the end of the rebuttal period.
>
> **"Synthesis models like WALT achieve forecasting performance on par with or better than models trained with mask-based objectives"**
>
> Table 1 on lines 324-330 touches upon this by replacing the forecasting diffusion model with a regression-based forecasting model as a deterministic, non-diffusion approach to synthesis. While WALT with regression is strong on most tasks, the 4DS features still outperform on box tracking. The 4DS representation was pre-trained with mask-based objectives, yet it still outperforms the diffusion based features of WALT. The use of other synthesis models (such as pure autoregressive approaches like VQVAE) is an interesting question and would be a basis for future work. However, this is not a trivial experiment to run. Most autoregressive approaches discretize the output space. For continuous latent spaces, it is unclear if alternate generative models (such as VAEs, Normalizing Flows) are mature enough to model these video datasets with high data diversity. We thus chose diffusion models which have an established track record on continuous output spaces with complex datasets.
>
> **Diffusion models are supposed to take different noise levels as input, with high noise levels capturing broad semantics and low noise levels capturing high fidelity details.**
>
> For the experiments in the paper, we selected a fixed low noise level. Although not included in the initial submission, we have conducted additional experiments to evaluate performance across a wide range of noise levels (from low to high). The results indicate that N-WALT achieves superior performance across all forecasting tasks at low noise levels. This observation aligns with findings in the paper “From image to video: An empirical study of diffusion representations”,  Vélez et al., 2025, which reports that WALT similarly excels in perception tasks when using low-noise levels. We will incorporate these experimental results of varying the noise level into the revised version of the paper in the rebuttal period.
>
> **"Language supervision does not result in better forecasting"**
>
> We are grateful for highlighting this oversight. We indeed meant vision-language contrastive training and will correct the language in the  revised version of the paper in the rebuttal period.
>
> **"Video backbones outperform image ones"**
>
> It is indeed true that the predictor model in DINO-world attends to past image frames to predict the future. In fact, similar to us, the DINO-world paper also claims that the temporal model can be left to this predictor. Like our diffusion model, they also use past temporal context to forecast the future. However, DINO-world claims that the base visual representation on top of which this temporally-aware predictor operates can simply be image-based, but we demonstrate experimentally  that base models trained on video outperform those trained on images alone. That being said, we note that while video models with temporal modeling do tend to be stronger overall, there are exceptions, as DINO outperforms on depth forecasting. We will clarify our claim and discussion in the paper in the rebuttal period.
>
> **The authors also do not control for resolution of feature spaces or model params or pre-training data volume in their analysis.**
>
> This is a valid concern, as different model architectures naturally have different feature space resolutions (and thus potential information bottlenecks) as well as different pre-training datasets. It is difficult to control for all of these factors and will add this discussion in the updated submission in the rebuttal period.

---

### Official Review · Reviewer_tCqG · 2025-11-03

**Soundness:** 4
**Presentation:** 3
**Contribution:** 4
**Rating:** 8
**Confidence:** 3

**Summary:**

The paper introduces a unified way to test how frozen vision backbones forecast the future across four abstraction levels—pixels, depth, point tracks, and object boxes—by training lightweight readout heads for perception and a diffusion forecaster that predicts future latent trajectories (4 past → 12 future frames). Evaluation uses both per-example task metrics and dataset-level measures computed in each task’s output space and a variance check. Across nine backbones (image vs. video, contrastive vs. generative, with/without language), the authors find that forecasting generally correlates with perception quality, video-pretrained models outperform image-only, and language supervision offers no consistent gains.

**Strengths:**

1. Forecasting under uncertainty is central to video understanding. The paper cleanly articulates this gap and proposes a concrete, reusable protocol.
2. Four tasks spanning low→high-level structure offer a broad, apples-to-apples view.
3. The paper shows several insightful empirical findings, such as: forecasting strongly correlates with perceptual quality; language supervision offers little forecasting benefit; synthesis-trained WALT does especially well on pixel/depth FD. They are useful signals for the community.

**Weaknesses:**

1. Readout heads are trained on observed frames and then applied to forecasted latents at test time. If the forecaster induces a distribution shift in the latent space, readouts might underperform.
2. The approach is closely related to autoregressive video-generation models [1–3]. Adding a brief discussion situating this work within that line of research would improve clarity.

[1] ACDiT: Interpolating Autoregressive Conditional Modeling and Diffusion Transformer.

[2] Generative Pre-trained Autoregressive Diffusion Transformer.

[3] Self-Forcing: Bridging the Train-Test Gap in Autoregressive Video Diffusion.

**Questions:**

1. How sensitive are conclusions to the number of forecast samples per clip? Could some model rankings flip for larger N?
2. Can you share per-model training hours for the forecaster and readouts?

---

> ### Author Response · Authors · 2025-11-21
>
> We would like to thank you for your helpful comments. We appreciate that you found that our paper offers a “broad, apples-to-apples view”, that the subject of forecasting is “central to video understanding”, and the results demonstrate “several insightful empirical findings.” We address your comments below:
>
> **“'Readout heads are trained on observed frames and then applied to forecasted latents at test time. If the forecaster induces a distribution shift in the latent space, readouts might underperform.”**
>
> We acknowledge this is a potential risk for our approach, and we touch upon it in the limitations sections of the appendix on lines 734 (We will move this to the main paper): “Third, forecasting on frozen perception features may impose representational mismatches, especially for tasks requiring fine-grained temporal dynamics.” That being said, since we use the same diffusion-based forecaster for all evaluations of all base models, we believe that the comparison across models is still as close as we can get to an apples-to-apples comparison because any potential distributional bias in the forecaster would similarly affect all models.
>
>  While we don't observe this issue in practice, we do assuage some of these risks by constraining aspects of the latent distribution with layer normalization, forcing it to only take on values between 0 and 1. This way, we know at least, by construction, that the forecast and latent space share the same range of values.
>
> **'The approach is closely related to autoregressive video-generation models [1–3]. Adding a brief discussion situating this work within that line of research would improve clarity.**
>
> The papers you mention are a variation of diffusion models on video that are autoregressive in time. [1] is autoregressive on spatial-temporal blocks and leverages diffusion to generate unseen blocks. [2] and [3] are autoregressive on past video frames and use diffusion to forecast the latents of future frames, similar to architecture in our work. These approaches to autoregressive diffusion are potential design choices within our framework, and including them into our diffusion-based approach would be an interesting follow-up work. We will update the submission with these citations and this discussion.
>
> **How sensitive are conclusions to the number of forecast samples per clip? Could some model rankings flip for larger N?**
>
> We will run experiments to ablate the number of samples per clip and update the submission with results by the end of the rebuttal period.
>
> **Can you share per-model training hours for the forecaster and readouts?**
>
> The Forecaster and readout heads are trained in the same experiment (with stop gradients, not end-to-end), ranging 8-48 hours depending on the model and task. We will add a table of these numbers to the submission by the end of the rebuttal period.

---

> ### Author Response · Authors · 2025-12-03
> **N Samples Ablation**
>
> Here is an ablation of forecast samples per clip for the 4DS-e model, ranging from 5-15 samples. We find that, apart from improving the best of N metric (which would be expected), we do not see that results improve dramatically as the number of samples increase.
>
> | Model | Pixels Mean $\uparrow$ | Pixels Best $\uparrow$ | Pixels FD $\downarrow$ | Depth Mean $\downarrow$ | Depth Best $\downarrow$ | Depth FD |
> | :--- | :---: | :---: | :---: | :---: | :---: | :---: |
> | 4DS-e 5 Samples | 20.09 | 21.76 | 31.74 | 0.193 | 0.119 | 538.54 |
> | 4DS-e 10 Samples | 19.89 | 22.03 | 30.95 | 0.194 | 0.096 | 533.00 |
> | 4DS-e 15 Samples | 20.02 | 22.36 | 29.85 | 0.194 | 0.087 | 538.24 |

---

> ### Author Response · Authors · 2025-12-03
> **Compute Time**
>
> Here is a table of training time for various models and tasks with the associated TPU architecture used. In these experiments, the readout head and diffusion model are trained in parallel (but not end-to-end, there is a stop gradient applied). We find that there is a great deal of variability in training time between models and tasks, likely due to variation in model parameters, latent space resolution, and the resolution of the final output space.
>
> | Model | Pixels (ScanNet) | Depth (ScanNet) | Points (Perc. Test) | Boxes (Waymo) |
> | :--- | :---: | :---: | :---: | :---: |
> | VideoMAEv1 | 6.35\*\* | 5.98\*\* | 2.34\*\*\* | 9.13† |
> | VideoMAEv2 | 23.85\*\* | 23.35\*\* | 6.33‡ | 6.17‡ |
> | DinoV2 | 12.83\*\* | 12.48\*\* | 16.62† | 16.55† |
> | VideoPrism | 14.90\*\* | 13.85\*\* | 3.98‡ | 3.86‡ |
> | SigLip | 21.74\*\* | 21.56\*\* | 18.62† | 8.01‡ |
> | VJEPA | 31.08\*\* | 31.05\*\* | 9.19† | 9.13† |
> | 4DS-H | 23.25\*\* | 22.38\*\* | 5.05§ | 4.04§ |
> | 4DS-e | 45.41\*\* | 45.46\*\* | 13.14† | 13.11† |
> | Walt | 45.75\* | 45.77\* | 13.71† | 13.76† |
> | Native Walt | 9.78\*\* | 8.99\*\* | 9.63† | 9.00\*\* |
>
> **Experiment training time in hours.** Both readout head and diffusion model were trained in parallel, but not end-to-end (stop gradient). Models were trained on TPUs. \* represents 16 v5p chips, \*\* represents 4 v5p, \*\*\* 6 v5p, † 4 v6e, ‡ 16 v6e, and § 32 v5p.

---

### Author Response · Authors · 2025-12-03
**Rebuttal Executive Summary**

Dear Area Chair and Reviewers,

We would like to thank the reviewers for their time and helpful comments. Our paper proposes a novel unified evaluation framework of forecasting across a range of visual abstractions that enables, for the first time, an apples-to-apples comparison of frozen pretrained visual representations. We apply this framework to multiple state of the art pretrained image and video representations to investigate their forecasting performance.. We note that the reviewers found that our paper is “central to video understanding”, showing many “several insightful empirical findings.” In addition, they noted it to be a “novel evaluation framework” and the framework provides “clear empirical guidance for building high-performance visual forecasting systems.” We note that reviewers requested various ablative experiments and some more experimental details in order to better understand the approach. We have added results of these ablations in an updated manuscript and supplementary material (highlighted in red), in addition to posting directly here. In addition, reviewers requested some clarifications and justifications of claims made in the paper; we have also included them in the updated manuscript and added them as posts here.

**Justification of Claims (Reviewer JPXb)**
There was concern that some of our claims needed further clarification and justification. We responded to these concerns and incorporated them into the manuscript.

- Image vs Video:
There was a comment that our claim that video-based architectures outperform image-based was incorrect and that the DINO-world citation was counter to this claim. We clarify that DINO-world is using an image-based representation (DINO) but delegates all the temporal modeling to the forecasting model. In our case, we experiment with underlying representations that have already been pre-trained with temporal modeling (VideoMAE) separate from any temporal modeling in the diffusion model, and find that they outperform image-based models..

- Correlation of Perception and Forecasting:
Another issue was raised was that the correlation between perception and forecasting was a trivial and obvious finding. We clarified that while this is a general result, the underlying findings are more nuanced, and this correlation starts to break down at the higher end of performing models. As SOTA video models improve to be able to hold longer temporal context (current SOTA is only 16 frames!), we expect this correlation to break down even further.

- Language Supervision:
There was a comment on the claim that language supervision did not necessarily lead to better forecasting. We clarify in the manuscript that this was  to be models trained only via vision-language contrastive. Naturally, WALT received language supervision as well as temporal supervision as it is a video synthesis model conditioned on language and this claim does not apply to it.

**Effect of the Forecasting Model (Reviewer JPXb, 432F)**
Reviewers wanted to see if there was any potential bias due to the choice of a diffusion-based forecasting model (such as giving an advantage to a diffusion-based representation like WALT). We point to Table 1, an experiment that compares regression-based forecasting against diffusion, as one way to disentangle this variable and find similar trends across both approaches. We note that diffusion-based forecasting is superior to a regression-based one, since it does not assume only a single possible future is correct. That being said, this difference between diffusion and regression applies similarly to all models under consideration and therefore does not introduce additional bias to the comparison.

**Discussion of Limitations (Reviewer JPXb, X1vx):**
Originally our limitations section was erroneously in the supplementary. We moved it to the main paper and added a discussion regarding different model parameter sizes, latent space resolution, and different pre-training datasets, as kindly suggested by the reviewers.

---

> ### Author Response · Authors · 2025-12-03
>
> **Ablations (Reviewer 432F, tCqG, JPXb)**
>
> There were requests for some additional experiments to better understand various aspects of the framework. We have run them and added them to the manuscript:
> - Ablation of Readout Head Capacity (432F)
> Changing the readout head performance does not seem to change performance drastically
> - Ablation of Number of Samples from the Diffusion Model (tCqG)
> Apart from improving the best-of-N metric, which would be expected, the effect of increasing the number of samples is negligible.
>
> - Noise Levels (JPXb)
> We performed a sweep across a range of noise levels for NWALT and  confirmed our original setting by demonstrating that maintaining low noise levels is critical for
> achieving the best results across tasks.
>
> - Compute Time  (tCqG)
> A table of compute time for the main experiments across models has been added to the manuscript. We find that there is significant variation in the compute time required for training between different downstream tasks and frozen representations, as expected from the nature of the different base models and the different tasks and dataset sizes.
>
> - Image or Depth Prediction Methods as a Comparison  (tCqG)
> We are running a comparison with DINO-Foresight on the Scannet dataset. However, as this dataset is much larger than the Cityscapes dataset in the original setup of DINO-Foresight, this experiment is still running and will not finish by the end of the rebuttal period. We will update the camera ready submission with this result.

---

### Meta-Review · Area_Chair_Xgss · 2026-01-07

**Summary:**

This submission presents a novel framework for evaluating frozen vision models on forecasting tasks, utilizing latent diffusion models to predict future representations and employing distributional metrics for performance evaluation. The study benchmarks several pre-trained vision models and provides empirical insights into the relationship between forecasting ability and perception, as well as the surprising performance of video-pretrained models like WALT.

The authors’ rebuttal effectively addresses several concerns, such as clarifying the use of language supervision, providing additional experiments on the readout head, and offering justification for the use of forecasting as a proxy for visual model quality.

However, some concerns, such as dataset bias and the impact of the diffusion model, were not sufficiently addressed in the rebuttal. Given the major revisions required, I would not object to recommending acceptance. I encourage the authors to make further revisions to address these issues and consider submitting to a future venue after ensuring that the concerns are fully resolved.

**Reviewer Concerns:**

- Most concerns related to methodological clarification (e.g., the correlation between forecasting and perception, language supervision, and model comparisons) have been partially or fully addressed by the authors, with additional experiments or explanations provided in the revised manuscript.

- Concerns related to the lack of detailed technical analysis, such as mathematical formulations, experiments with non-diffusion synthesis models, and deeper analysis of architectural choices, have not been sufficiently addressed. These remain significant gaps in the paper.

- Concerns about dataset bias and computational details (e.g., GPU architecture, training hours) have been partially acknowledged, but the authors need to provide further details to fully resolve these issues.

**Reviewer Scores:**

While the paper has made progress in responding to reviewer feedback, it still requires more technical depth and additional experimental comparisons to fully address all concerns and potentially change the minds of reviewers who gave lower scores (including Reviewer 432F, who rated it a 6). Further revisions would help clarify the contributions and methods, making them more rigorous and convincing.

---

### Decision · Program_Chairs · 2026-01-26

Reject